# Chemical Scalpel: An Experimental Collagenase-Based Treatment for Peritoneal Adhesions

**DOI:** 10.3390/biology11081159

**Published:** 2022-08-02

**Authors:** Javier Barambio, Mariano García-Arranz, Pedro Villarejo Campos, Juan Felipe Vélez Pinto, Luz Vega Clemente, Soledad García Gómez-Heras, Héctor Guadalajara, Damián García-Olmo

**Affiliations:** 1Fundación Jiménez Díaz University Hospital, 28033 Madrid, Spain; mariano.garcia@quironsalud.es (M.G.-A.); pedro.villarejo@quironsalud.es (P.V.C.); felipevelezpinto@gmail.com (J.F.V.P.); luz.vega@quironsalud.es (L.V.C.); h.guadalajara@quironsalud.es (H.G.); damian.garcia@uam.es (D.G.-O.); 2Department of Surgery, Faculty of Medicine, Universidad Autónoma de Madrid, 28029 Madrid, Spain; 3Department of Human Histology, Faculty of Health Sciences, Rey Juan Carlos University, 28922 Madrid, Spain; soledad.garcia@urjc.es

**Keywords:** collagenase, adherences, enzymatic adhesiolysis, peritoneum treatments, peritoneal surgery, experimental model

## Abstract

**Simple Summary:**

Intraperitoneal adhesions are bands of scar tissue that occur frequently after abdominal surgery. These scars bind the abdominal organs together, producing symptoms such as intestinal obstruction, infertility, chronic pain, and a greater risk of injury in subsequent surgery. Currently, the only treatment approach for this disease is a risky surgical intervention that may cause additional adhesions or other complications. In this article we propose the use of collagenase in the peritoneal cavity to facilitate adhesion disruption. Using a simple experimental rat model, we produced an array of adhesions resembling those presented in humans. We demonstrate that the application of collagenase at the concentration and time described is safe and facilitates the disruption of adhesions with no organ damage due to contact with collagenase. The further development of this therapy and application route, published for the first time in this article, may improve the quality of life of patients with this disease.

**Abstract:**

(1) Background: Abdominal adhesions are a common disease appearing after any type of abdominal surgery and may prolong surgical time and cause intestinal obstruction, infertility, or chronic pain. We propose the use of intraperitoneal collagenase to perform chemical adhesiolysis based on the pathophysiology and histology of adhesions. (2) Methods: We generated an adhesion model with intraperitoneal polypropylene meshes. Four months later, we evaluated the efficacy of the treatment in blinded form, i.e., 0.05% collagenase vs. placebo at 37 °C for 20 min. Protocol 1: Ten rats with ten mesh fragments, in which an attempt was made to remove the maximum number of meshes in a 5-min period. Protocol 2: Six rats with four mesh fragments in the sides of the abdominal cavity in which adhesiolysis was performed using a device that measures burst pressure. (3) Results: Protocol 1: 42% efficacy in the collagenase group versus 8% in the control group (*p* < 0.013). Protocol 2: 188.25 mmHg (SD 69.65) in the collagenase group vs. 325.76 mmHg (SD 50.25) in the control group (*p* < 0.001). (4) Conclusions: Collagenase allows for the safe and effective chemical adhesiolysis in this experimental model of adhesions.

## 1. Introduction

Intraperitoneal adhesions are a normal response to inflammation of the peritoneal surfaces following surgery or acute abdominal diseases. These adhesions are found in 95% of previously operated patients who undergo subsequent surgery [1,2], and can cause significant morbidity such as bowel obstruction, female infertility, chronic abdominal pain, and increased complications and duration of subsequent surgery [3,4,5,6]. The annual cost burden of adhesion-related complications in the U.S.A. is estimated at over 2 billion U.S. dollars [7].

Adhesions are the most common cause of intestinal obstruction in Western countries [8,9,10]. Patients with a history of abdominal surgery involving the peritoneum may be at greater risk of developing peritoneal adhesions with each intervention, and adhesion severity may increase with each successive procedure. In the Ellis cohort [11], which included 21,347 surgical readmissions, 5.7% were classified as being directly related to adhesions and 3.8% required adhesion-related reoperation. 

Current treatments are mainly preventive. Minimally invasive surgery is recommended to reduce peritoneal injury, avoid extensive surface coagulation, and optimize hemostasis [12], as these factors can alter fibrinolytic activity [3,13,14]. Promoting intestinal transit by means of prokinetics and active visceral mobilization is a proven preventive measure [15,16,17]. Few pharmacologic agents are safe for intraperitoneal administration in humans, and even fewer have empirical evidence justifying their use. Those currently in use aim to prevent contact between peritoneal surfaces for the first five to seven days; however, such lack of contact may impair the healing of intestinal anastomoses by increasing the incidence of anastomotic leakage. In terms of physical barriers, the use of hyaluronic acid films is the only approach shown to reduce intestinal obstruction [1,18,19], although this intervention has significant side effects in colorectal surgery [20]. 

This study aims to evaluate a collagen-based enzymatic approach developed to inhibit the formation of peritoneal adhesions following abdominal surgery. Collagenase is a promising candidate which, during its short half-life (between 6 and 30 min) [21], acts as an enzymatic debridement agent capable of hydrolyzing collagen peptide bonds.

To date, collagenase has only been used clinically as a standard treatment for benign conditions such as Dupuytren’s disease, Peyronie’s disease, as well as for enzymatic debridement of dermal ulcers [22,23]. However, the drug has been extensively evaluated in in-vitro studies, showing that, when used at adequate concentrations, it does not affect cell viability. One report on the use of collagenase in animal models showed that when administered intratumorally or intravenously, low doses of collagenase delivered for more than 4 h has an impact on tumors but becomes toxic, causing lethal abdominal and pulmonary hemorrhage in murine models [24], and as a result, controlling enzyme exposure time poses a challenge. The same authors have shown that a single high dose of collagenase (700 U/mL or 4.5 mg/mL) also produces toxicity, whereas a dose of 37.5 U/mL (0.24 mg/mL) caused no organ toxicity and no presence of the enzyme in the blood.

Over the last 4 years, our research group has conducted a series of studies that aimed to verify the safety of this procedure, placing emphasis on time of exposure and enzyme concentration. As we previously found that the drug can be used safely at concentrations of 37 U/mL for periods of less than 45 min, this report presents data related to the use of collagenase for the treatment of post-surgical peritoneal adhesions.

### Preliminary Results (Toxicity Test)

Animals treated at concentrations of 350 U/mL (2.24 mg/mL) died after failing to recover from anesthesia at an average of 60 min postoperatively. Necropsy showed hemorrhage in the mesentery of the small intestine and tissue fragility. Despite the overdosage, laboratory testing was a challenge, as collagenase is not detected in blood by enzyme-linked immunosorbent assay (ELISA). Animals with concentrations above 70 U/mL show peritoneal tissue fragility. However, in order to ensure the absence of toxic effects caused by the use of collagenase in the peritoneal cavity, we decided to use a dose of 37.5 U/mL for 30 min in all trials, since laboratory and histologic results can be extrapolated to the control group.

## 2. Materials and Methods

### 2.1. Legal and Ethical Considerations

This study was approved by the Clinical Experimentation Ethics Committee of the Hospital Universitario Fundación Jiménez Díaz (no. PIC/75-2016). The procedure was approved by the OEBA (animal welfare agency) of the region of Madrid with reference number PROEX 226/17. All experiments were conducted in accordance with national and international regulations on the protection of experimental animals. 

### 2.2. Animals

Twenty six-week-old female Wistar rats weighing between 100 g and 200 g were used for the experiments. The animals were bred in accordance with the 2010/63/EU directive and had unrestricted access to water and standard rat chow. Environmental conditions (light, ambient temperature, and relative humidity) were maintained homogeneously. All surgeries were performed with the animals under anesthesia and following the administration of antibiotic prophylaxis (cefazoline 30 mg/Kg). The animals received subcutaneous analgesic treatment for the first 5 postoperative days (tramadol 5 mg/kg every 24 h).

### 2.3. Step 1: Development of an Adhesion Model

Two different adhesion models (models A and B) were generated. Both were created by fitting the rats with high-density polypropylene meshes (Parietene™) of two different sizes and placements depending on the adhesion model:

In model A (N = 10), 10 meshes of 1 mm^2^ were evenly distributed on both sides of the abdomen in each animal. In model B (N = 6), four meshes of 2 × 4 mm^2^ were placed in each rat, with two on each flank. A control group of four rats receiving no intervention was assembled (Figure 1).

Once inserted, the meshes were kept in place for four months to allow the adhesions to consolidate. The second surgery was performed to treat the adhesions and the third to evaluate the state of the peritoneum after treatment, with four months between each procedure. At the time of the second and third surgeries, we defined four possibilities according to the findings (Figure 2): Type 0 (no adhesions found); Type 1 (one or two adhesions between surfaces); Type 2 (>2 adhesions between surfaces); Type 3 (deep mesenteric adhesions); and Type 4 (bowel obstruction).

### 2.4. Step 2 (+4 Months): Treatment of Adhesions

The procedures were blinded in such a way that the surgeon did not know whether he was using the treatment solution (collagenase in PBS) or the control solution (phosphate saline solution, PBS). The peritoneal treatments compared in the two models were as follows: administration of 20 mL of solution containing 37.5 U/mL of collagenase in 0.05% PBS for 20 min in the treatment group versus 20 mL of PBS solution for 20 min in the control group. In both, a final lavage was performed with Ringer’s lactate. All solutions were tempered to 37 °C prior to use.

Previously, tests had been performed to verify the safety of the collagenase used (Mix Type I and Type II, Lyposmol S.L., Madrid, Spain) at a given concentration when applied intraperitoneally in a rat model. For this purpose, a solution containing collagenase in PBS at different concentrations was applied by vigorous washing of the peritoneal surface at 37 °C for 20 min [25].

#### 2.4.1. Protocol A (Adhesion Model A)

Ten rats were randomized into two study groups. The second surgery was performed in blinded fashion. During this intervention, the surgeon attempted to remove as many mesh fragments as possible in a 5-min period. We used a scale to measure the difficulty of mesh removal: 1 = loose mesh; 2 = mesh release with light or blunt traction and no bleeding; 3 = release with significant traction, bleeding, or need for fastening elements; 4 = release requiring sharp instruments or risk of organ injury. Extractions requiring only type 1 or type 2 operations were considered successful.

#### 2.4.2. Protocol B (Adhesion Model B)

Six rats were randomized into two study groups. The treatment was performed in a blinded manner. Subsequently, traction was applied, measuring adhesion burst pressure (ABP). The number of tractions was determined by the exposure of the tissue after treatment.

For protocol B, a system was designed to measure the ABP. It consists of a high-precision manometer connected to a pressure pump and a low-resistance plunger that allows the adhesion tissue to be clamped. This system is fitted on a support box where the animals are placed for evaluation. In this way, we quantified the maximum pressure required to separate the adhesion. The operation was repeated once for each adhesion exposed (Figure 3).

### 2.5. Step 3 (+8 Months): Final Revision Surgery

Revision surgery was performed four months after the second surgery. In this step, we described the adhesions found and obtained samples of different abdominal tissues (peritoneum and muscle). The animals were euthanized at the conclusion of the surgery.

### 2.6. Monitoring and Tests Performed

We recorded the weight of all animals in the treatment and control groups over time. Serial analyses performed on the 20 animals (pre-surgery, immediately before the treatment, 24 h after treatment, and at sacrifice) included complete blood count and biochemistry tests. After the treatment phase, we obtained peripheral blood samples to measure circulating collagenase by ELISA so as to determine levels of MMP1 and MMP2 (Abnova GmbH; Aachen, Germany treatment in both study groups and 15 days after the intervention, using Luminex using the Bio-Plex Pro™ Rat Cytokine 23-Plex Assay (Ref. M60009RDPD, Bio-Rad, Madrid, Spain); 50 µL of plasma was used from each animal and time point according to the manufacturer’s protocol. All assays were carried out in triplicate.

### 2.7. Histological Studies

After the animals were euthanized, 5-mm^3^ tissue samples from the abdominal organs were fixed in 10% formaldehyde at room temperature, embedded in paraffin, and cut into 5-micron-thick slices in a Micron HM360 microtome. To evaluate the peritoneal adhesions and inflammation around the tissues, the histological sections were stained with hematoxylin-eosin. All were studied under a Zeiss Axiophot 2 microscope and photographed with an AxiocamHRc camera. The evaluating pathologist was blinded to the type of intervention performed on each animal.

### 2.8. Statistical Analysis

Data were analyzed with IBM SPSS version 22 and graphs were assembled with Microsoft Excel. A one-way ANOVA test was performed for independent nonparametric samples, followed by Tukey’s post-hoc comparison tests (type of adhesions generated, difficulty of mesh removal, growth curve, laboratory findings). A student’s *t*-test was used for parametric tests. Results with a *p*-value < 0.05 were considered statistically significant.

## 3. Results

### 3.1. Monitoring and Tests Performed

All animals survived both interventions. There were no wound complications or other incidents concerning the maintenance of the rats. The average weight gain in the collagenase group was 33.67% after the first surgery and 30.99% after the second. The mean weight gain in the control group was 33.16% after the first surgery and 23.50% after the second. The weight increase within the untreated group was 48.48% and 25.31%, respectively. Differences in final weight and in the growth curve were not significant (*p* = 0.91).

No statistically significant differences between groups were found for any of the parameters studied, i.e., biochemistry, blood count, and coagulation. The results of circulating cytokine analysis revealed differences (Table 1) over time, which will be commented on in the discussion.

### 3.2. Histopathological Results

In the control group, the findings of the histological studies performed at sacrifice show adhesions (fibrous scar tissue) between the abdominal wall and the bowel (Figure 4A,C) and also extensive areas of chronic inflammation on the wall.

The histopathological results of the abdominal organs between the collagenase-treated group and the placebo-treated group are very similar to each other. (Here we focus on the abdominal wall and small intestine.) In these organs, after collagenase was administered, the extracellular matrix of the adipose tissue around the abdominal wall and small bowel was decreased and adipocytes were separated from each other (Figure 4(B1,B2)). The external wall of the abdominal viscera was still intact (Figure 4(D1), red arrow), whereas the collagen fibers of the adhesion were dispersed due to the action of collagenase (Figure 4(D2)).

### 3.3. Adhesion Development

The adhesions observed after the creation of the adhesion model and after treatment vs. placebo surgery are shown in Figure 5 and Figure 6. The differences between groups were not statistically significant (experiment A, *p* = 0.264; experiment B, *p* = 0.513).

### 3.4. Experiment A

No differences were found in the type of adhesions after the first surgery (*p* = 0.264) or after the second surgery (*p* = 0.446). With respect to success, defined as extraction difficulty type 1 (free mesh) or 2 (slight traction without bleeding), 21/50 meshes were successfully extracted in the treatment group (42%), while only 4/50 meshes (8%) were extracted in the placebo group (*p* = 0.013).

### 3.5. Experiment B

No differences were found in the type of adhesions after the first (*p* = 0.513) or second surgery (*p* = 0.189).

The number of tractions performed in each case depends on the number of adhesions exposed after lavage. Seventeen tractions were performed in the placebo group and 27 in the treatment group (*p* = 0.08).

The mean traction pressure used was 325.76 mmHg in the placebo group (SD 50.25) and 188.25 mmHg (SD 69.65) (*p* < 0.001) in the treatment group (Figure 7).

## 4. Discussion

The peritoneum of the animals studied showed a similar clinical response to polypropylene mesh application, as seen in the human peritoneum. An aseptic inflammatory reaction is produced which generates firm adhesions in four months. As in humans, the final stage of adhesive syndrome in the murine model is intestinal obstruction. For this reason, we believe that the adhesion model is useful for testing new therapies.

The results from these first phases of safety and fine-tuning of the models show that the treatment is safe and effective at the concentrations used. Furthermore, we observed no serious side effects or mortality associated with the use of collagenase; similarly, histologic and laboratory studies did not reveal any differences. No lesions were identified in the organs and tissues in contact with collagenase in the histopathological study. In no case was collagenase detected in circulating blood.

The present study does not confirm that collagenase prevents adhesions, as this was not the objective of the study. However, intestinal obstruction (type 4) only appeared in three animals treated with placebo. Moreover, the growth curves after the second surgery were slightly higher in the group treated with collagenase (30.99%) vs. placebo (23.50%). These differences did not reach statistical significance (*p* = 0.91).

Cytokine/chemokine analysis shows that treatment with collagenase generates a greater increase in MCP-1 and VEGP proteins, which indicates the activation of platelets and type 2 macrophages [26]; although we did not detect collagenase in the blood, this could indicate slight lesions in the wall of the abdominal veins, though this was not detected macroscopically or histologically. In addition, collagenase treatment generated a greater increase in the cytokines IL2 and IL18, which are involved in the increased secretion of macrophages and T-lymphocytes [3,27]. The modest increase in TNF alpha appears to indicate a slight increase in cell apoptosis, a finding consistent with the disruption of the intercellular stroma by collagenase. Finally, the increase in IL12p70 was not associated with an increase in IFNgamma, which leads us to believe that the results at 15 days could be associated with an early-phase chronic inflammatory process that could be summarized as activation of type 2 macrophages and T-lymphocytes and no activation of mononuclear or polynuclear cells involved in the early resolution of an inflammatory process.

The experiments demonstrated that washing with intraperitoneal collagenase facilitates the removal of polypropylene meshes in a statistically significant way compared to washing with the diluting agent.

In the first experiment, we observed a higher number of successful extractions due to the greater exposure of the peritoneal tissues adhered. In addition, when traction was performed, there were minimal lesions to adjacent tissues with the use of collagenase. In the second experiment, we observed that the pressure used to disrupt the adhesions was significantly lower in the group treated with collagenase compared to the group receiving placebo. The number of traction points achieved was greater in all the individuals in this group, indicating that the exposure of the adhered peritoneum is better with collagenase treatment.

Foreign body reaction is normal around a mesh that has been implanted for months [28,29]. Intraperitoneal adhesions are seen as fibrous scar tissue. It appears that collagenase breaks down this tissue and, incidentally, the inflammatory foci. The abdominal walls are therefore almost normal. 

The approach to peritoneal adhesions or other peritoneal diseases using proteolytic enzymes applied directly to the serous membrane could be effective and opens up a new line of research that may help solve important and widespread clinical problems.

## 5. Conclusions

When applied at a certain concentration, temperature, and volume, collagenase is useful in releasing peritoneal adhesions caused in reaction to polypropylene meshes.

No morbidity or mortality was reported during the treatment. The growth curves and laboratory and histologic studies performed over time in the treated models can be extrapolated to those of the placebo-treated controls.

We consider the peritoneal pathway and treatment with collagenase to be a useful and effective approach that merits further research.

## 6. Patents

Authors J. Barambio, D. Garcia-Olmo, M. Garcia-Arranz, H. Guadalajara, and L. Vega-Clemente have applied for one patent related titled “Chemical Scalpel” (EP 19382118.8).

## Figures and Tables

**Figure 1 biology-11-01159-f001:**
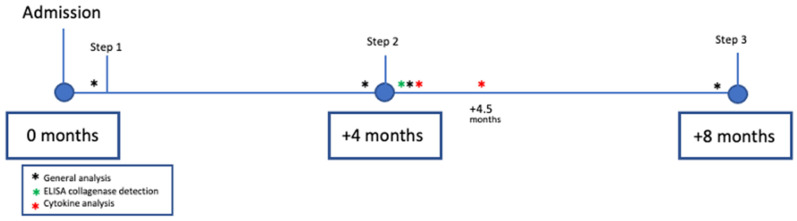
Flow chart: Step 1: (+1 month): Adhesion generation; Step 2: (+4 months): Adhesion analysis and treatment. Mesh removal after treatment; Step 3: (+8 months): Analysis of newly generated adhesions associated with the surgical process and necropsy of animals.

**Figure 2 biology-11-01159-f002:**
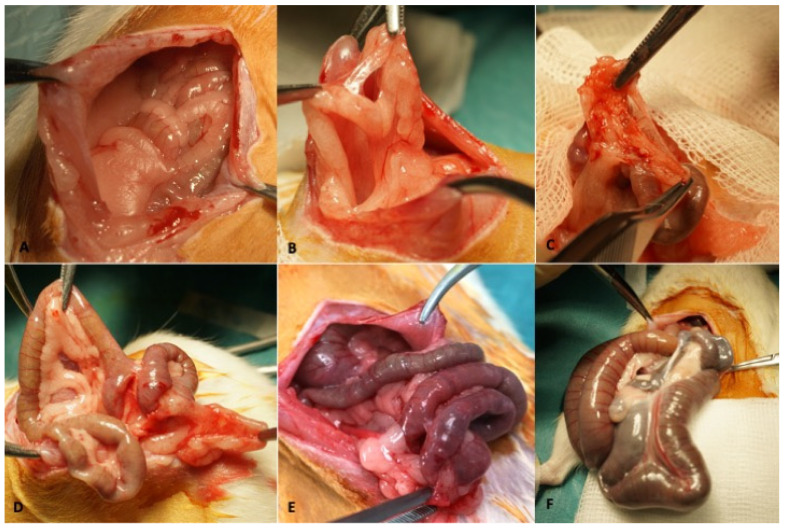
Type of adhesions. (**A**): Type 0; (**B**): Type 1; (**C**,**D**): Type 2; (**E**): Type 3; (**F**): Type.

**Figure 3 biology-11-01159-f003:**
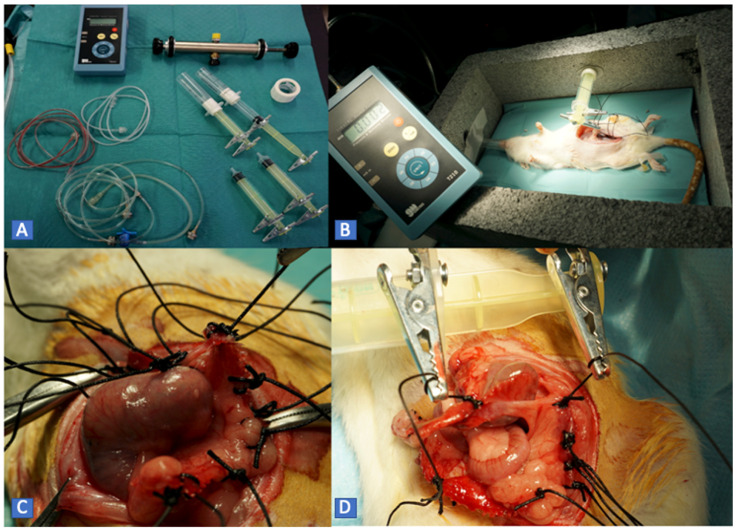
(**A**) Elements of the traction and measurement system before assembly. (**B**) System with its corresponding box, which allows traction to be performed without interfering with the elements or the model. (**C**) Detail of the placement of the traction sutures. (**D**) Close-up of adhesiolysis within the traction system.

**Figure 4 biology-11-01159-f004:**
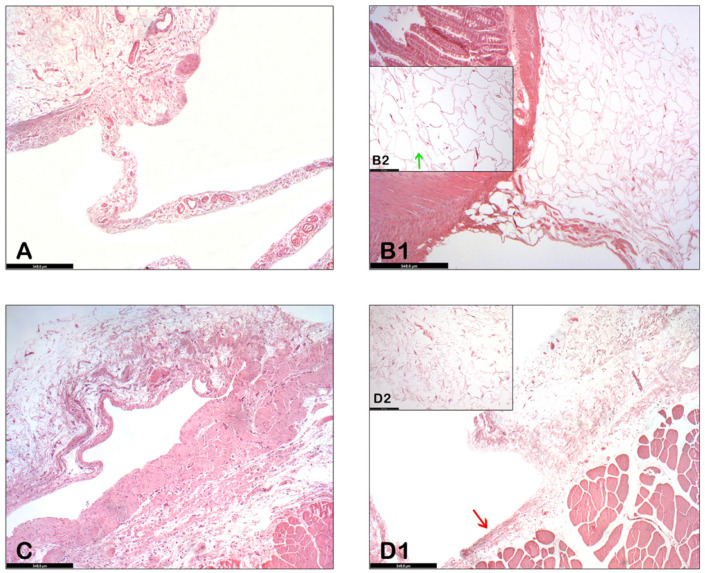
(**A**). Adhesion arising from the small bowel wall in the non-treatment group; (**B1**). Similar intestinal adhesion after collagenase exposure (hematoxylin-eosin staining, original magnification ×50). (**B2**). The adipocyte matrix is degraded and the cells are separating from each other (hematoxylin-eosin staining, original magnification ×200); (**C**). End of adhesion in the abdominal wall in the non-treatment group (hematoxylin-eosin staining, original magnification ×50); (**D1**). Abdominal wall in the collagenase-treated group, visceral walls remain intact after administration of collagenase (red arrow) (hematoxylin-eosin staining, original magnification ×50); (**D2**). collagen fibers dispersed due to the action of collagenase (hematoxylin-eosin staining, original magnification ×200).

**Figure 5 biology-11-01159-f005:**
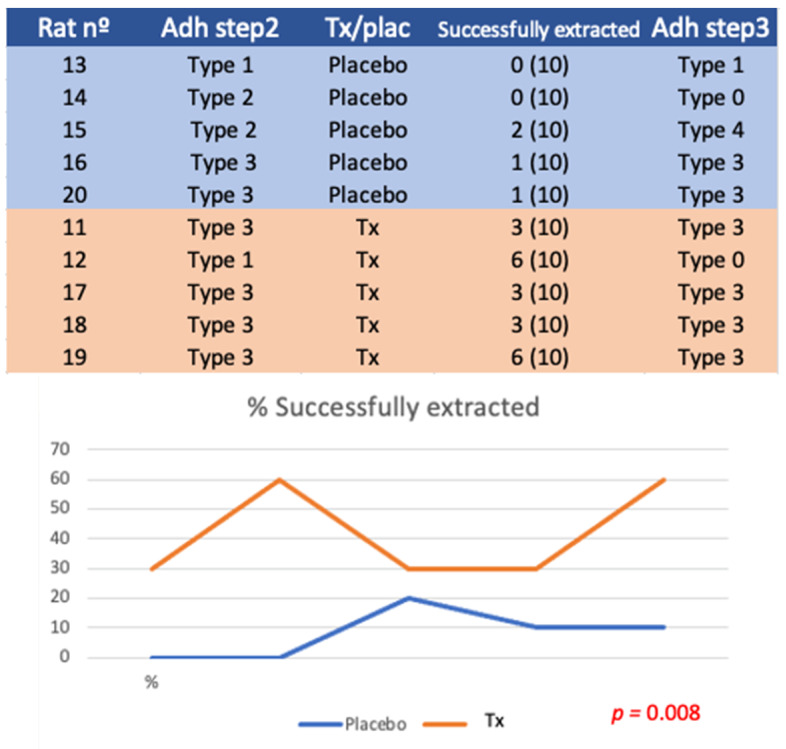
Table experiment A. Successful extraction by type 1 or type 2 maneuver. Graph of experiment A. Differences in the number of meshes obtained with treatment vs. placebo are represented in % (*p* = 0.013).

**Figure 6 biology-11-01159-f006:**
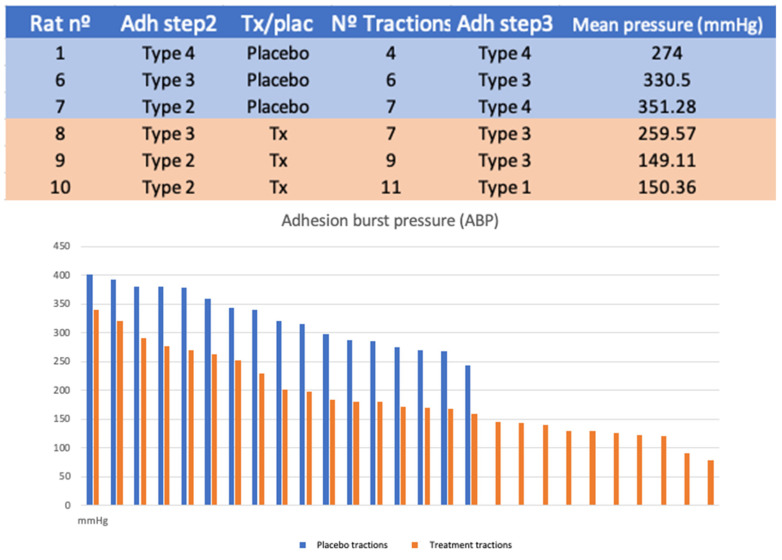
Table experiment B. Graph of experiment B: pressure distribution according to the tractions performed (*p* < 0.001).

**Figure 7 biology-11-01159-f007:**
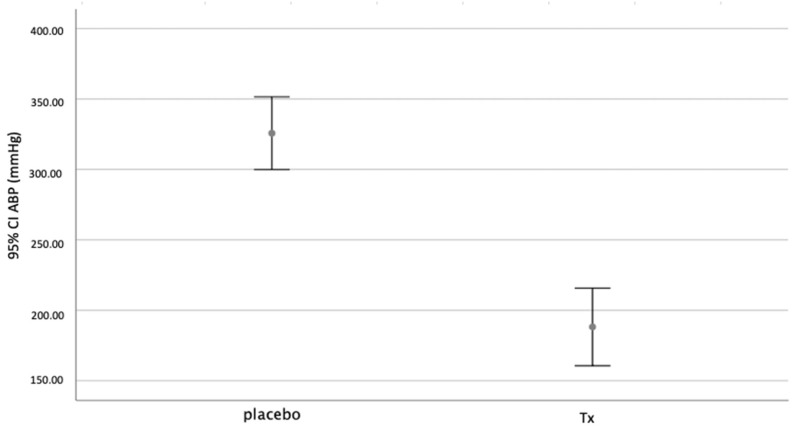
Mean traction pressure. *p* = 0.001.

**Table 1 biology-11-01159-t001:** Results of circulating cytokine analysis.

	Control	1 Day Post PBS Tx	15 Days Post PBS Tx	1 Day Post Collagenase Tx	15 Days Post Collagenase Tx
IL 1a	387.96	1654.1	786.35	1929.18	644.53
IL 1b	421.66	1585.96	1254.77	2178.82	950.36
IL 2	5654.96	15,215.62	7166.43	19,992.78	5463.02
IL 5	758.94	1746.12	1068.87	2073.42	967.41
IL 6	1373.81	4790.76	198.93	5579.95	2222.29
IL 7	410.35	1576.80	1283.37	2103.03	954.97
IL 13	418.51	1710.88	613.40	2195.32	735.11
IL 17a	92.74	301.86	139.56	396.99	13,920.00
IL 18	5820.69	14,830.24	8337.55	18,712.95	7670.80
M-MCF	67.16	465.88	152.90	397.23	121.96
MIP 3a	74.45	170.75	101.07	197.54	71.56
GM-CSF	260.87	1011.85	789.70	1313.72	620.63
VEGF	702.43	2204.43	1041.05	2801.54	1027.1
MCP 1	1437.90	3305.61	2219.33	4136.09	1968.94
G-CSF	24.03	56.21	18.66	87.62	23.11
RANTES	235.39	502.26	233.17	641.06	283.23
KC	154.91	392.10	233.34	507.47	209.85
MIP 1a	54.41	200.53	168.83	283.38	129.27
IL-4	38.57	810.58	332.11	1022.56	350.97
IL-10	86.02	1098.21	521.75	1399.38	465.26
INF-µ	900.93	3646.19	1304.22	4669.56	1551.85
TNF α	1172.64	4689.04	1452.15	6555.31	2077.08
IL12 p70	1394.55	4732.25	1830.98	6389.92	1964.08

The mean of inflammatory cytokines and chemokines (pg/mL) in the different groups and times from plasma.

## Data Availability

Not applicable.

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
