# Peer review of "Chemical Scalpel: An Experimental Collagenase-Based Treatment for Peritoneal Adhesions"

_biology, 2022, doi:10.3390/biology11081159_

Round 1
Reviewer 1 Report
They generated the adhesion model to proposed the collagenase could prevent adhesion.
Yes, its an interesting work. Compared with other research work, this work is relatively simple.
The written was fine and the text is clear and easy to read.
The conclusions are consistent with the evidence and arguments is also fine.
Yes, they addressed the main question.
But, the authors need add the mechanism of preventing adhesion after operation was tested by cell experiment.
Author Response
We would like to thank the constructive comments and suggestions of the reviewer. We have answered each of your points below.
Q1: They generated the adhesion model to proposed the collagenase could prevent adhesion.
Response 1: The adhesion model generated is based on the inflammatory reaction secondary to the contact of the peritoneum with the polypropylene mesh. After testing different models, we have concluded that this procedure is the most similar to adhesions in humans, as well as being easy to reproduce. For this reason, we consider it suitable to carry out the experiments.
The aim of the study is to demonstrate that the use of collagenase facilitates subsequent adhesiolysis. Apparently, the animals treated with collagenase develop fewer adhesions, but since this is not the objective of the study, we cannot conclude that it prevents them. We will perform further studies with this objective.
Q2: Yes, it’s an interesting work. Compared with other research work, this work is relatively simple. The written was fine and the text is clear and easy to read. The conclusions are consistent with the evidence and arguments is also fine. The conclusions are consistent with the evidence and arguments are also fine.
Response 2: Thank you very much for your comment.
Q3: Yes, they addressed the main question. But, the authors need to add if the mechanism of preventing adhesion after operation was tested by cell experiment.
Response 3: Thank you very much for your positive comment. We reviewed the different types of peritoneal collagen published previously and type I collagen (80%) is mainly involved in peritoneal healing and it is diffusely distributed in the peritoneal stroma [1, 2]. To break down type I collagen, we used a commercial product composed of a mixture of type I and II collagenase (Lyposmol ADSCenzyme (Lyposmol s.l., Spain) from Clostridium histolyticum (ref N0002936). Previously, in our laboratory we have used this collagenase to release mesenchymal stem cells from the fatty stroma, and we used it to break “in vitro” tumor stroma from biopsies of human abdominal cancers. In both assays we observed a rupture of the extracellular matrix and release of the cells into the culture dishes. However, control of temperature, collagenase concentration and time are essential to avoid damaging the cells, similar to how it is done in our rat model. Analyzing this “in vitro” assays we observed that our collagenase broke type I and III collagen during 40 minutes, at 37ºC, or less time depending on collagen concentration in culture dish.
- (Witz, C.A., Montoya-Rodriguez, I.A., Cho, S. et al. Composition of the Extracellular Matrix of the Peritoneum. Reprod. Sci. 2001. 8, 299–304. https://doi.org/10.1177/107155760100800508).
- (Bittinger F, Schepp C, Brochhausen C, Lehr HA, Otto M, Köhler H, Skarke C, Walgenbach S, Kirkpatrick CJ. Remodeling of peritoneal-like structures by mesothelial cells: its role in peritoneal healing. J Surg Res. 1999;82(1):28-33. doi: 10.1006/jsre.1998.5449. PMID: 10068522).
Reviewer 2 Report
Peritoneal adhesion is a common problem with different complications, and there is not any proper treatment for this problem, so performing more studies is need for finding about pathological pathway an effective therapeutic methods.
Author Response
We would like to thank the reviewer for his constructive comment.
Q1: Peritoneal adhesion is a common problem with different complications, and there is not any proper treatment for this problem, so performing more studies is needed to learn about pathological pathway effective therapeutic methods.
Response: Thank you very much for your comment. In addition to research, our main activity is of a clinical nature. We are a team of surgeons, in collaboration with a research laboratory, who treat patients with this unresolved problem on a daily basis. We believe that our work can be a first step to help patients with this disease and develop an effective treatment in the future.
Reviewer 3 Report
I am honored of the review invitation. Based my expertise in the field of bioengineering and biomaterials, I would judge this paper as Accepted with Minor Reversion.
The research group described a feasible strategy, which collagenase was applied as the adhesiolysis regent, to prevent the pathological adhesion formation. The experimental evidence is solid and conclusion is clear. However, I would recommend an insight into following questions.
1. Animal model validity. Please provide detailed illustration on the simulation of the rat mesh graft model to the clinical scenarios where the main trigger of postoperative adhesion is the local tissue injury.
2. Profile of biosafety. While research group gained solid evidence in the systematic biocompatibility of collagenase, local immune response and tissue remodeling are absent. Please provide the profile of acute and chronic immune cells infiltration into adhesion tissue, as well as the remodeling changes of collagen and fibronectin during the peritoneum healing.
3. Cohort evaluation. This paper highlighted that all the rats survived and intestinal obstruction occurred in the placebo group. However, to my limited animal work experience, these two fact seems conflict since rat highly likely to die after developing intestinal obstruction, especially in long term. Please explain.
Overall, after polishing the essential and foundational questions above, I believe this paper is qualified to be published by the Journal of Biology.
